# Glucosinolates and Isothiocyantes in Processed Rapeseed Determined by HPLC-DAD-qTOF

**DOI:** 10.3390/plants10112548

**Published:** 2021-11-22

**Authors:** Ana Miklavčič Višnjevec, Angelica Tamayo Tenorio, Anne Christine Steenkjær Hastrup, Natanya Majbritt Louie Hansen, Kelly Peeters, Matthew Schwarzkopf

**Affiliations:** 1Faculty of Mathematics, Natural Sciences and Information Technologies, University of Primorska, Glagoljaška 8, 6000 Koper, Slovenia; kelly.peeters@innorenew.eu (K.P.); matthew.schwarzkopf@innorenew.eu (M.S.); 2Center for Bioresources, AgroTech, Danish Technological Institute Gregersensvej, 2630 Taastrup, Denmark; ante@dti.dk (A.T.T.); acha@teknologisk.dk (A.C.S.H.); nmlh@teknologisk.dk (N.M.L.H.); 3InnoRenew CoE, Livade 6, 6310 Izola, Slovenia

**Keywords:** glucosinolates, isothiocyanates, HPLC-MS/MS, rapeseed processing, protein powder

## Abstract

Glucosinolates are well known as natural antimicrobials and anticarcinogenic agents. However, these compounds can lose their properties and transform into antinutrients, depending on processing conditions. In addition, the bitterness of some glucosinolate in rapeseed meal can affect the likability of the final product. Therefore, it is important to identify and determine each glucosinolate and its derived form, not just the total glucosinolate content, in order to evaluate the potential of the final rapeseed protein product. This study provides a comprehensive report of the types and quantities of glucosinolates and their derived forms (isothiocyanates) associated with different rapeseed processing conditions. Glucosinolates and isothiocyanates were determined by HPLC-DAD-qTOF. In our study, the enzymatic degradation of glucosinolates by myrosinase was the main factor affecting either glucosinolate or isothiocyanate content. Other factors such as pH seemed to influence the concentration and the presence of glucosinolates. In addition, process parameters, such as extraction time and separation technology, seemed to affect the amount and type of isothiocyanates in the final protein extracts. Overall, both determined intact glucosinolates and their derived forms of isothiocyanates can give different types of biological effects. More studies should be performed to evaluate the impact of glucosinolates and isothiocyanates on human health.

## 1. Introduction

Glucosinolates are natural sulphur-containing compounds that are produced by the secondary metabolism of plants from the order Capparales, which among other families includes the Brassicacea. Several of these *Brassica* species are important for feed ingredients, notably rapeseed protein, whereas other species are commonly used in human nutrition, such as cauliflower, cabbages, broccoli and Brussels sprouts. Glucosinolates and their derived products determine the typical flavour and (bitter) taste of these vegetables and feedstock [1]. According to Wallsgrove et al. [2], more than 100 different glucosinolates have been identified. The general structure of glucosinolates is characterized by a SCN group, a sulphate, and a β-D-glucopyranosyl. According to the chemical structure of the side chain (-R), among others, alkenyl-, phenyl- and indolylmethyl glucosinolates can be distinguished [3]. These compounds are involved in plant defense mechanisms against depredation, and some of these compounds appear to be protective against chemical carcinogens [2]. Glucosinolates are naturally converted into isothiocyanates by enzymes in the plants. Isothiocyanates contain the chemical group –N=C=S, formed by substituting the oxygen in the isocyanate group with a sulphur. Glucosinolates can also be converted to the aglycone by enzymatic hydrolysis with myrosinase [4]. Depending on the substrate, pH and availability of ferrous ions, glucosinolates can decompose into isothiocyanates, thiocyanates or nitriles. At physiological pH levels, isothiocyanates are the major product. Glucosinolates and the enzyme myrosinase are segregated in intact plant tissues [5]. When the compartmentalization of glucosinolates and myrosinase is lost by the plant, physical damage that can occur after freezing and thawing, chopping and the myrosinase-catalyzed metabolism of glucosinolates occurs [6,7,8,9]. Isothiocyanates formed by hydrolysis of glucosinolates may act as chemoprotective agents against carcinogens and have a positive effect on cardiovascular and neurological diseases [10]. Moreover, isothiocyanates may act as inhibitors of mitosis in tumour cells [11,12,13].

Glucosinolates and their derived forms are detected and identified by high-performance liquid chromatography with tandem mass spectrometry (HPLC-MS/MS). All intact glucosinolates contain at least two sulphur atoms and, therefore, yield a characteristic molecular ion isotope pattern. This aids in confirmation of the presence of sulphur atoms and, therefore, of glucosinolates [14]. The glucosinolates characteristic fragment ions [SO_4_]^−^, and [SO_4_H]^−^ (*m*/*z* 96, 97) are formed with high abundance in the high and programmed cone voltage ESI method. The simultaneous occurrence of both *m*/*z* 96 and 97 ions is a highly diagnostic fingerprint for glucosinolates, even when these compounds are present in complex mixtures [14]. Although the abundance ratios were not constant for all glucosinolates examined by Mellon et al. [14], they all fell within the range 1:2 to 1:4 ([*m*/*z* 96]/[*m*/*z* 97]) [14]. Therefore, the identities of the glucosinolate peaks can be then established by extracting the full-scan spectra at the retention times indicated by the increase in abundance of the *m*/*z* 96 and 97. The mass spectra of HPLC peaks that lacked the 96, 97 pair of ions can also be checked to determine whether these are glucosinolates. Mellon et al. [14] showed that in all cases HPLC peaks lacking the 96, 97 pair of ions were proven not to be glucosinolates. Regarding isothiocyanates, high-sensitivity detection and quantification by LC-MS/MS can be achieved by derivatization with ammonia to the corresponding thiourea derivative and positive ion electrospray ionization LC-MS/MS. Thiourea derivatives can be detected by LC-MS/MS with multiple reaction monitoring (MRM) transitions that generally involve a loss of ammonia [4].

The detection and identification of glucosinolates and their derivatives are of importance during the processing of raw materials such as rapeseed cake, due to the concentration steps towards protein products, where glucosinolates or isothiocyanates can accumulate and may reach undesirable levels. Rapeseed meal and pressed cake are mostly by-products from the oil extraction processing of rapeseed. These meals/cakes contain between 30–40% high-quality protein and have been the subject of considerable attention. Rapeseed meal produced from hot pressing, followed by hexane extraction, results in substantial denaturation of proteins due to excessive heat exposure, rendering a proportion insoluble in water and, therefore, more difficult to isolate using aqueous extraction processing. Moreover, rapeseed meals and cakes contain glucosinolates and degradation products that impart bitter tastes, and phytate (inositol hexa-phosphate), which are all anti-nutritional compounds and limit the utilization of rapeseed meals/cakes. Nevertheless, processing technologies are being developed to ensure a low glucosinolate content of products that can be an alternative source of protein for various food and feed applications.

With the current pressure on protein prices, increasing demand for functional ingredients, and remaining controversy over wider use of soy, there is an excellent case for fractionating rapeseed residues after oil extraction to produce protein concentrates and isolates while removing glucosinolates and phytates [15,16]. This study examines glucosinolate and isothiocyanate content of rapeseed protein products obtained from different processing conditions, using hot and cold pressed rapeseed cake as raw material. To our knowledge, this is the first study to provide a comprehensive report of the types and quantities of glucosinolates and their derived forms (isothiocyanates) associated with different rapeseed processing conditions towards final protein products.

## 2. Results and Discussion

### 2.1. Glucosinolates in Raw Material

Table 1 shows the results for glucosinolates determined in cold-pressed rapeseed cake (CPR), listing the name of the identified compound, its molecular formula, molar mass, *m*/*z*, retention time (RT), characteristic fragments and the 96/97 ratios. In addition, the concentrations of the most important glucosinolates such as progoitrin, gluconapin, sinigrin, sinalbin and gluconasturtiin were determined using appropriate analytical standards.

As shown in Table 1, progoitrin, sinigrin, sinalbin, gluconapoliferin, glucoalyssin, gluconapin, 2-methylpropyl glucosinolate or butyl glucosinolate, glucobrassicanapin, 4-methyltiobutyl glucosinolate, gluconasturtiin, three isomers of methylpentyl glucosinolate and neoglucobrassicin were tentatively identified in CPR. Progoitrin, sinigrin, sinalbin, gluconapin and gluconasturtiin were confirmed by analytical standards. No distinction was possible between isomers such as 2-methylpropyl glucosinolate or butyl glucosinolate having the same molecular formulas. The identified glucosinolates in CPR are in accordance with the literature [17,18,19,20]. The most commonly identified glucosinolates in rapeseed in the literature [17,18,19,20] are gluconapin, glucobrassicanapin, progoitrin, gluconapoliferin, glucobrassicin and gluconasturtiin. These compounds were also identified in our study, except for glucobrassicin. Glucobrassicin could be present in our samples at concentrations below the measuring concentration range, as has been previously reported by Heaney [17]. From different studies [17,18,19,20], including the current one, it is evident that the glucosinolate profile differs between the same types of rapeseed cake, probably due to genetic and environmental factors.

Within different groups divided according to different anabolic pathways of glucosinolates, the most abundant were aliphatic glucosinolates such as progoitrin, sinigrin, gluconapoliferin, glucoalyssin, gluconapin, etc. Aliphatic glucosinolates are compounds mainly derived from methionine, but also valine alanine, leucine and isoleucine. This group of glucosinolates showed the strongest inverse association with cancer risk. At a more precise level, glucosinolates and their degradation products have been shown to reduce the risk of several cancers, such as colon [21,22], bladder [23], lung [24], breast [25] and prostate [26]. Dietary glucosinolates have been reported to block the formation of endogenous or exogenous carcinogens to prevent the onset of carcinogenesis [27]. They are indirect antioxidants because they do not neutralize free radicals directly, but do so by modulating xenobiotic metabolic enzymes (phase I and phase II enzymes) [28]. However, it must be pointed out that health risks and health benefits associated with the consumption of glucosinolates in the glucosinolate-derived ingredient are defined by structure, including the stereochemistry of food compounds, and their dietary concentrations. Both intact glucosinolates and their metabolites can give different types of biological effects [29] including hypothyroidism and enlargement of the thyroid gland in mammalian species [1]. Glucosinolates have been shown to be mutagenic and weakly genotoxic [30,31,32,33].

The main quantified compounds present in cold-pressed rapeseed cake (CPR) were gluconapin that amounts to 2413 ± 27 mg/kg and progoitrin that amounts to 1014 ± 56 mg/kg. The same compounds were some of the most abundant in rapeseed according to Millan et al. [20]. They reported 120–2233 mg/kg in the case of progoitrin and 93–966 mg/kg in the case of gluconapin. The exception was 4-hydroxyglucobrassicin that was not identified in our study and was present in the range of 139.4–2133 mg/kg according to Millan et al. [20]. However, Sosulski et al. [19], did not identify this compound. They determined neoglucobrassicanapin (4350 mg/kg), progoitrin (1550 mg/kg) and gluconapin (970 mg/kg) as the most abundant in rapeseed meal. In contrast to Sosulski et al. [19], neoglucobrasicanapin in CPR was tentatively identified near the limit of detection in MS in our study, wheres Millan et al. [20] did not identify this compound. The total quantified glucosinolates were near the lowest determined values in most of the cases, but in the ranges according to the literature [1,19,20,34]. Differences in the amount of glucosinolates in *Brassica* plants are attributed to genetic and environmental factors, including plant age, temperature, water stress and soil type [35,36].

The determined concentrations in the raw material (CPR) showed that the selected sample exhibited results similar to plant varieties with low glucosinolate content. The measured values were more than 50% lower compared with the limit value of 20 mmol/kg seed dry matter for plants with low glucosinolate content [1].

### 2.2. Glucosinolates during the Processing of the Pressed Rapeseed Cake

Rapeseed cake coming from hot or cold pressing processes was submitted to different processing phases to obtain a final protein powder. All included samples went through (1) extraction, (2) separation, (3) concentration and (4) spray drying. However, different process conditions (e.g., solid to water ratio, separation technologies) were tested in the various trials during pilot-scale process developmental. Figure 1 shows the whole process of extraction, separation and concentration of CPR, and glucosinolate and isothiocyanate content in samples taken at different stages. Glucosinolates were only found in Sample S1 (aqueous extract before soaking), containing a smaller amount of progoitrin and gluconapin, and neoglucobrassicin identified at the limit of detection (0.3 mg/kg dry weight). These results could be due to enzymatic degradation of glucosinolates by myrosinase, resulting in a number of secondary reaction products such as isothiocyanates, nitriles, thiocyanates, epithionitriles, or oxazolidin-2-thiones [37]. Isothiocyanates were found in samples S3–S8 and they were expected to be formed due to the processing conditions used. The decomposition of glucosinolates is influenced by different factors such as pH. At pH 5–7, isothiocyanates are usually formed, whereas at slightly lower pH values, nitriles are usually formed [38]. Native conditions (pH~5) were used during the processing of CPR in this study. Besides degradation of glucosinolates by myrosinase, hydrolysis is also expected during the aqueous extraction process [1].

### 2.3. Glucosinolates in Different Extracted Rapeseed Protein Sample

In the protein products from trials carried out with CPR (Trials CPR-1; CPR-2; CPR-3; and CPR-4), the glucosinolate content was below the limit of detection (LOD). However, different glucosinolates were found in protein products from trials carried out with hot-pressed rapeseed cake (HPR) (Trials HPR-1; HPR-2; and HPR-3 (Appendix A). The highest peaks in all three samples corresponded to progoitrin and gluconapin (Table 2). In the case of HPR, the processing temperatures exceed 70 °C during oil pressing. Therefore, the enzyme myrosinase is likely to have less than 22% of its activity [39]. In this raw material, most of the glucosinolates are still present.

Ten different glucosinolates were identified in the extracted rapeseed protein extract from Trial HPR-3, namely progoitrin, gluconapoliferin, glucoalyssin, gluconapin, 2-methylpropyl glucosinolate/butyl glucosinolate, glucobrassicanapin, gluconasturtiin, neoglucobrassicin, 3-methylpentyl glucosinolates and 4-methyltiobutyl glucosinolate. All the glucosinolates that were found in Trial HPR-3 were identified in Trial HPR-2, with the exception of 4-methyltiobutyl glucosinolate (Table 2). In Trial HPR-1, only five glucosinolates were identified. Progoitrin, gluconapoliferin, gluconapin and neoglucobrassicin were identified with lower concentrations compared with other samples, whereas butyl glucosinolate/2-methylpropyl glucosinolate was identified with approximately the same concentrations as in Trial HPR-2. The differences in glucosinolate content between the samples might be due to the different processing parameters tested such as enzymatic extraction at different pH conditions, reaction time and filtration technologies (see the Materials and Methods). For instance, pH was adjusted to pH 8 in Trial HPR-1, pH 6.7 in Trial HPR-2 and pH 5, followed by pH 7 in Trial HPR-3, suggesting that basic medium might affect the stability and conversion of glucosinolates. Jing et al. [40] observed a diminished glucosinolate content after adjusting the pH from 8 to 9, whereas the content of glucosinolates changed only slightly between the pH range from 3 to 8.

Table 2 shows the quantification results of the identified glucosinolates. Highest concentrations of determined glucosinolates were found for progoitrin and gluconapin, and the highest levels were found in Trial HPR-3 with a total amount of 4972 ± 348 mg/kg (13.0 ± 0.9 mmol/kg). About 20% lower concentrations were found in Trial HPR-2 (3917 ± 274 mg/kg; 10.1 ± 0.7 mmol/kg) and about 40% lower concentrations were found in Trial HPR-1 (3129 ± 219 mg/kg; 8.1 ± 0.8 mmol/kg) compared with Trial HPR-3. The high levels observed in Trial HPR-3 exceed the EFSA recommended restriction of total glucosinolates level, which amounts to 1–1.5 mmol/kg for monogastric animals [1]. However, data on the toxicity of individual glucosinolates for food-producing animal species are very limited. In addition, the extracted rapeseed protein is meant for human consumption, and reliable data on limited recommended intakes for human consumption are rather difficult to obtain. In literature, only rough estimates are found for the recommended limit values, which amount to 0.25 µmol/kg body weight/day for the Netherlands, 0.5 to 0.6 µmol/kg body weight/day for Canada and the United States of America and 1.6 µmol/kg body weight/kg body weight for the United Kingdom [41]. According to the lowest (Netherlands) and the highest (United Kingdom) recommended value, for a person of 70 kg, the consumption of rapeseed final protein product should not exceed 1.3–8.6 g/day from Trial HPR-3, 1.7–11.1 g/day from Trial HPR-2 and 2.2–14 g/day from Trial HPR-1. More studies should be carried out to evaluate the impact of the identified glucosinolates on human health at expected lower concentrations.

### 2.4. Isothiocyanates in Raw Material and during Processing of Rapeseed Pressed Cake

Isothiocyanates were determined in the raw material and during the processing of the cold-pressed rapeseed cake (Figure 1). In the CPR raw material, only 4-hydroxybutyl isothiocyanate was found at the limit of detection. The lack of isothiocyanates in the raw material can be explained by a temporary low activity of the enzyme myrosinase in CPR, whereas the CPR it is in its dry form. Since this raw material has only been treated at mild temperatures (<70 °C), once water is added for extraction, the enzyme myrosinase is expected to be active and able to catalyze the cleavage of glucosinolates to aglycons that further decompose into isothiocyanates, thiocyanates, nitriles or epithionitriles. Figure 1 shows the concentrations of the main compound goitrin, which is present at the different stages of pilot-scale rapeseed protein extraction for Trial CPR-1. Goitrin loss during processing of rapeseed was assessed to the amount of 669 ± 50 mg/kg wet weight. This compound is derived from 2-hydroxy-3-butenyl isothiocyanate formed by hydrolysis of progoitrin, which was one of the main compounds in the raw material. Goitrin was also the main compound found in samples from other Trials with CPR (Trials CPR-2, CPR-3 and CPR-4; see Table 3 and Appendix A).

In addition to goitrin, 2-phenylethyl isothiocyanate was found at the limit of detection in the remaining solids after the aqueous extraction in Trial CPR-1 (Appendix A). That process stream is discarded, thereby eliminating 2-phenylethyl isothiocyanate from the process. In addition, small intensities of alyssin (the conversional product of glucoallysin) that were below limit of quantification were still present in S4, the liquid fraction that continues the process, but this compound was not detected in the subsequent samples; 2-phenylethyl isothiocyanate, the conversional product of gluconasturtiin, was present in the final protein product from Trial CPR-3 (Table 3) at the limits of detection; 4-methylthiobutyil isothiocyanate, the conversional product of 4-methylthiobutyl glucosinolate was detected in the raw material and the final protein product in Trial CPR-2, although at the limit of the detection (Table 3); 3-butenyl isothiocyanate, the conversional product of gluconapin, was only found in the final protein product in Trial CPR-4, but below the limit of quantification. The differences in isothiocyanates detected in samples from different trials should be due to different processing conditions (See Materials and Methods), as was previously reported by Deng et al. [42]. Especially at the beginning of the processing, after the solid/liquid separation, different filtration technologies were applied that could affect the accumulation/elimination of the converted isothiocyanates. For instance, the highest content of goitrin was determined in the protein extract from Trial CPR-2. In this trial, goitrin had twice the concentration compared with Trial CPR-1 (Table 3). The concentration was approximately 13 times higher compared with Trial CPR-3 and approximately 4 times higher compared with Trial CPR-4. This might be due to the filtration procedure (i.e., 0.2 µ microfiltration followed by 5 kDa ultrafiltration) used in trial CPR-2, compared with trial CPR-1 (i.e., manual liquid/liquid separation followed by 5 kDa ultrafiltration) and trial CPR-3 (i.e., 0.2 µ microfiltration followed by 10 kDa ultrafiltration) suggesting that accumulation of goitrin can occur under specific filtration conditions. Moreover, trial CPR-4 used the same filtration condition but included a second aqueous extraction step, which might have affected the glucosinolate conversion. Another process parameter that differs among these trials is the duration: 5-day process for trial CPR-2 vs. 4-day process for trials CPR-1, CPR-3 and CPR-4. The additional extraction day seemed to have affected the glucosinolates and isothiocyanates. Finally, goitrin might be the most abundant of the isothiocyanates also due to its stability and could be used as a biomarker of glucosinolate conversion to isothiocyanates in rapeseed and its products.

Isothiocyanates were also analyzed in samples from trials using hot-pressed rapeseed cake (HPR); however, none were detected. In this raw material, the enzyme myrosinase is expected to be completely inactivated during oil pressing, due to heat inactivation. HPR processing involves temperatures above 70 °C, which is higher than the inactivation temperature of myrosinase [43]. Therefore, glucosinolates cannot be converted into isothiocyanates in HPR.

Isothiocyanates were determined in our study, not just because they can be conversional products of glucosinolates but also because they are potentially the most bioactive products of glucosinolate hydrolysis [44]. The most harmful degradation product detected in our study was oxazolidin-2-thione (goitrin) derived from progoitrin [35]. The parent compound (progoitrin) accumulates in rapeseed and causes goiter and other adverse effects on animal health, such as depressed growth, poor chicken egg production and liver damage [45]. Goitrin is an inhibitor of thyroid peroxidase and prevents the oxidation of iodide to iodine for subsequent iodination of tyrosine residues in T3 and T4 thyroxine biosynthesis. Thiocyanate anions act as a competitive iodide inhibitor, thereby preventing iodide uptake into the thyroid [46]. Langer et al. [47] reported that 25 mg (194 µmol) was the minimal amount of goitrin that decries the uptake of radioiodine. According to the results obtained in this study (Table 3), this amount is reached by consuming 0.26 kg CPR-2, 3.57 kg CPR-3 and 1.14 kg CRP-4. Moreover, high levels of progoitrin found in HPR-1, HPR-2 and HPR-3 can be also converted in goitrin. In this case the amount would be reached by consuming only grams of final protein products. The excesses consumption of these protein products should be avoided. However, there was no evidence of any effect of goitrin on humans due to its ingestion [45]. In addition, it has been shown that goitrin can cause a nitrosation reaction that can affect human health when nitrate levels in water are high [46]. The exact cause of this is not fully understood, but it may be related either to the presence of intact glucosinolates or to the production of nitriles in the digestive tract [18].

On the other hand, studies have shown that progoitrin, epiprogoitrin and their derivatives (goitrin and epigoitrin) are thought to exhibit a number of biological activities, including antiviral, anti-inflammatory, and anti-tumour effects [48,49]. Additional studies should be performed in order to fully evaluate isothiocyanates bioactive effects in humans and animals exposed at low doses.

## 3. Materials and Methods

### 3.1. Sample Description

All samples tested (e.g., HPR-2 Trials) for this study were obtained from the Biorefinery Pilot Plant of Danish Technological Institute (DTI), Taastrup, Denmark.

In CPR cake, the pressing occurs at slightly lower temperatures than HPR since no external heating is involved during pressing [50]. However, this does not mean that the process occurs at low temperatures (>10 °C) because friction inside the pressing equipment creates some heat, and temperatures as high 70 °C can be reached.

HPR cake, as described by the Pro-Enrich supplier Emmelev, is produced as follows: rapeseeds are dried to about 5.5% water content, followed by sieving, milling, pressing, transfer to a cascade mixer at minimum 81 °C and final separation and cooling of the rapeseed cake before storage and later shipment. During pressing, the machine is heated to improve oil release and the temperatures normally exceed 100 °C.

All the samples were stored at −18 °C prior to analysis.

#### Pilot-Scale Rapeseed Protein Extraction

All trials were performed under the same unit operations, notably: extraction process, separation and filtration, and spray drying. During the extraction, the rapeseed cakes at different solid-to-water ratios were subjected to various conditioning regimes (Figure 1, Table 4). A decanter centrifuge was employed for solid/liquid separation. Membrane filtration was used for liquid/liquid separation (i.e., 0.2 µm microfiltration) and concentration (i.e., 10 kDa ultrafiltration or 300 Da nanofiltration). No further filtration was carried out before analysis. The final protein extract was dried by spray drying, resulting in a fine powder. Liquid samples were taking from reactor or buffer tanks under continuous stirring. Solids samples were collected from (1) the decanter/centrifuge outlet after all the material was separated, aiming at a homogeneous sample, and (2) from the spray drier at the product outlet. Either the liquid or the dry form of the protein extract was used for glucosinolate content analysis. The liquid samples were freeze-dried (Alpha 1–4, Martin Christ Buchi) prior to analysis. The dry powder yield of the protein end products CPR-1 to CPR-4 and HPR–1 to HPR-3 is shown in Table 5. All trials were performed a single time as part of upscaling activities for industrial process development.

### 3.2. Determination of Glucosinolates

The extractions of glucosinolates in rapeseed meal and protein product samples were performed according to instructions by Tolra et al. [3], with some modifications. Certified reference material (CRM) ERM-BC190 rapeseed (S, total glucosinolate, medium level) was previously ground to a fine powder. The glucosinolates were extracted from 100 mg of the dry sample and CRM with 70% aqueous methanol (*v*/*v*) in a boiling water bath for 5 min. After cooling and centrifugation (4000 RPM, 5 min), the proteins of the supernatant were precipitated with 200 µL of a 0.1 M solution containing lead acetate and 200 µL of a 0.1 M solution containing barium acetate. After centrifugation (4000 RPM, 5 min), the supernatants were filtered through Whatman No. 4 paper. After that, samples were evaporated with a rotavap at 35 °C to remove methanol and water. Water was added to the obtained pellets and filtered through a 0.2 µm/PA (Nylon) filter before being analysed by HPLC-ESI-qTOF, as was described by Torla et al. [3]. CRM ERM-BC190 rapeseed was used to check the recovery of the extraction method, as was previously reported by Torla et al. [3], and the found value (21.4 ± 1.1 mmol/kg) was in good agreement with the certified values (23 mmol/kg ± 4 mmol/kg). The method rendered a 93 ± 7% recovery, and that was in agreement with previously reported data [3].

Glucosinolates were characterized using a high-pressure liquid chromatography system (HPLC), interfaced with a qTOF mass spectrometer (HPLC-ESI-QTOF/MS). The HPLC was equipped with a Poroshell 120 column (EC-C18; 2.7 µm; 3.0 × 150 mm). Elution was performed using mobile phase A (0.1% formic acid aqueous solution) and mobile phase B (0.1 formic Acid: acetonitrile, 50:50). The flow rate was 0.4 mL/min, and detection was at 229 nm. Two gradient systems were used were as follows: 2–10% B, 0–10 min; 10–50% B, 10–25 min; 50–90% B, 25–30 min; 20–30% B. The re-equilibration time was 5 min [51]. The separation compounds were first monitored using DAD (230 nm), and then MS scans were performed in the range *m*/*z* 40–1000, in the following conditions: capillary voltage, 2.5 kV; gas temperature 250 °C; drying gas 8 L/min; sheath gas temperature 375 °C; sheath gas flow 11 L/min. The MS acquisition was performed in negative ionisation mode as was reported by Mellon et al. [14]. In those conditions, the instruments were expected to provide experimental data with accuracy within ± 3 ppm. All data were processed using Qualitative Workflow B.08.00 and Qualitative Navigator B.080.00 software. The screening strategies were the same as reported elsewhere [4,14] (Appendix A). In addition, more specific characteristic ions 259.012 (C_6_H_11_SO_9_^−^) and 96.960 (HSO_4_^−^) in MS^2^ [52] were used for identification confirmation of the glucosinolates. Gluconapin potassium salt (SI-PHL89765-10MG, Sigma Aldrich, Merck KGaA, Darmstadt, Germany), progoitrin potassium salt (SI-PHL89765-10MG, Sigma Aldrich, Merck KGaA, Darmstadt, Germany), sinalbin potassium salt (SI-PHL89793-10MG, Sigma Aldrich, Merck KGaA, Darmstadt, Germany), gluconastuirtiin potassium salt (SI-PHL89689-10MG, Sigma Aldrich, Merck KGaA, Darmstadt, Germany) and sinigrin hidrat (SL-00290-10MG, Sigma Aldrich, Merck KGaA, Darmstadt, Germany) were used for quantification of glucosinolates. The calibration plots indicated good correlations between peak areas and commercial standard concentrations (Appendix A). Regression coefficients were higher than 0.990 (5 points per calibration graphs). LOD varied from 0.1 mg/kg to 0.37 mg/kg dried weight, whereas LOQ varied in the range from 1.0 mg/kg to 3.7 mg/kg dried sample. The standard deviation between duplicates was less than 7%.

### 3.3. Determination of Isothiocyanates

High-sensitivity detection and quantification of isothiocyanates by LC-MS/MS was achieved by derivatization with ammonia to the corresponding thiourea derivative and positive ion electrospray ionization LC-MS/MS. Thiourea derivatives were detected by LC-MS/MS with MRM transitions that generally involved a loss of ammonia [4]. For the derivatization procedure, 50–250 mg of powder sample (or 200 µL standard) was mixed with 2M ammonia in isopropanol (3 mL, 2 mL- std) and homogenized. The homogenate was left at room temperature for 24 h. Excess ammonia and isopropanol were evaporated using a rotavapor at 35 °C. To the obtained pellets, 1 mL (2 mL for std) of methanol was added and filtered through a 0.2 µm/PA (Nylon) filter before analysis by HPLC-ESI-qTOF [4].

Isothiocyanates were characterized using an HPLC-ESI-QTOF / MS. The HPLC was equipped with a Poroshell 120 column (EC-C18; 2.7 µm; 3.0 × 150 mm). The elution gradient of water/formic acid (99.05: 0.5, *v*/*v*) (A) and methanol (B) was performed as follows: with a linear gradient of 0 to 80% methanol from 0 to 20 min and isocratic 80% methanol from 20 to 35 min. The separation was performed at a flow rate of 0.5 mL/min, using 1 µL of injection volume. The separation compounds were first monitored using DAD (240 nm) and then MS scans were performed in the range *m*/*z* 40–1000, in the following conditions: capillary voltage, 2.5 kV; gas temperature 250 °C; drying gas 8 L/min; sheath gas temperature 375 °C; sheath gas flow 11 L/min. The MS acquisition was performed in positive ionisation mode as was reported by Song et al. [4]. In those conditions, the instruments were expected to provide experimental data with accuracy within ± 3 ppm. All data were processed using Qualitative Workflow B.08.00 and Qualitative Navigator B.080.00 software. The screening strategies were the same as reported elsewhere [4]. Goitrin (SI-PHL85696, Sigma Aldrich, Merck KGaA, Darmstadt, Germany) and 2-phenylethyl isothiocyanate (AL-253731-5G) were used for quantification. LOD varied from 0.07 mg/kg sample to 0.42 mg/kg sample, whereas LOQ varied from 0.7 mg/kg sample to 4.2 mg/kg dried sample. The standard deviation between duplicates was less than 7%.

## 4. Conclusions

In this study, glucosinolates and isothiocyanates were determined from different processing strategies for the extraction of protein from rapeseed cake. The enzymatic degradation of glucosinolates by myrosinase was the main factor in our study affecting glucosinolate and isothiocyanate content. In the case of hot-pressed rapeseed cake (HPR), temperatures higher than 70 °C were used during oil pressing, and, therefore, the enzyme myrosinase was deactivated, leaving intact glucosinolates in the samples during protein extraction. Thus, no isothiocyanates were determined due to the lack of conversion of glucosinolates by the enzyme. In the case of cold-pressed rapeseed cake (CPR), only isothiocyanates were detected in the final products since all glucosinolates were converted by myrosinase. This enzyme is presumed to be active in CPR due to the lower temperatures to which the rapeseeds are exposed during pressing. Other factors such as pH seemed to influence the concentration and the presence of glucosinolates. In basic media (pH = 8), the concentration of glucosinolates was lower compared with lower pH (near 7), probably due to the partial degradation of glucosinolates that can occur in alkaline pH conditions. In addition, process parameters, such as extraction time and separation technology, seemed to affect the amount and type of isothiocyanates in the final protein extracts. More studies should be performed to evaluate the impact of the identified glucosinolates on human health since both intact glucosinolates and their metabolites can give different types of biological effects.

## Figures and Tables

**Figure 1 plants-10-02548-f001:**
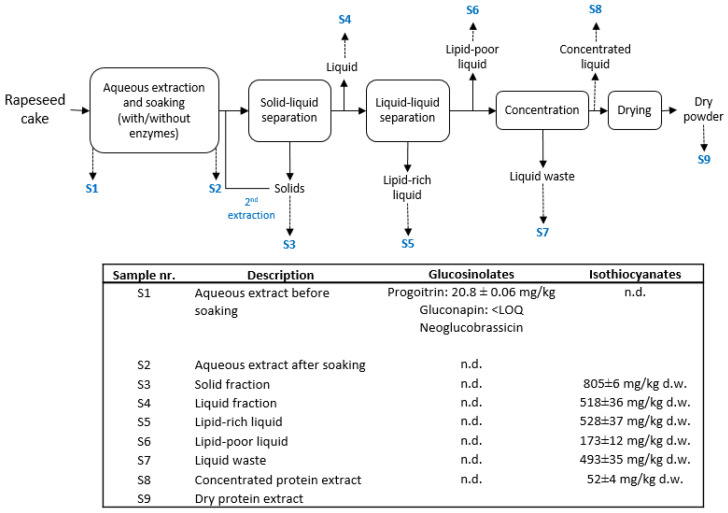
Glucosinolates and goitrin determined at different stages of pilot-scale protein extraction from cold-pressed rapeseed cake (CPR) during trial CPR-1. n.d.—not detected.

**Table 1 plants-10-02548-t001:** Results of glucosinolates in cold-pressed rapeseed cake (CPR).

Name of the Compound	Formula	M	[M − H]^−^	Characteristic Fragments	Fragment 96/97 Ratio	RT	Conc.
						(min)	
2-Hydroxy-3-butenyl glucosinolate (progoitrin)	C_11_H_19_NO_10_S_2_	389.0460	388.0389	96.9595	1: 2	2.26	1014 ± 56 mg/kg
				259.0183			(2.6 mmol/kg)
2-Propenyl glucosinolate (sinigrin)	C_10_H_17_NO_9_S_2_	359.0287	358.0287	96.9586	1: 2	2.92	3.6 ± 0.09
							mg/kg
				259.0012			(0.01 mmol/kg)
*p*-Hydroxybenzyl glucosinolate (sinalbin)	C_14_H_19_NO_10_S_2_	425.0460	424.0387	96.9595	1: 4	4.39	21.7 ± 2.2
				259.0106			mg/kg
							(0.05 mmol/kg)
2-Hydroxy-4-pentenyl glucosinolate (gluconapoliferin)	C_12_H_21_NO_10_S_2_	403.0620	402.0549	96.9602	1: 3	4.98	114.8 * ± 8.0
				259.0130			mg/kg
							(0.28 mmol/kg)
Glucoalyssin	C_13_H_25_NO_10_S_3_	451.0660	450.0586	96.9602	1: 2	5.51	52.1 * ± 0.04
				259.0079			mg/kg
							(0.12 mmol/kg)
3-Butenyl glucosinolate (gluconapin)	C_11_H_19_NO_9_S_2_	373.0510	372.0439	96.9601	1: 3	5.93	2413 ± 27
				259.0146			mg/kg
							(6.5 mmol/kg)
2-Methylpropyl glucosinolate/butyl glucosinolate	C_11_H_21_NO_9_S_2_	375.0670	374.0598	96.9523	1: 3	10.2	16.3 * ± 1.1
				259.0247			mg/kg
							(0.04 mmol/kg)
4-Pentenyl glucosinolate	C_12_H_21_NO_9_S_2_	387.0670	386.0598	96.9601	1: 3	12.3	23.7 * ± 1.7
(glucobrassicanapin)				259.0124			mg/kg
							(0.06 mmol/kg)
4-Methyltiobutyl glucosinolate	C_12_H_23_NO_9_S_3_	421.0540	420.0472	96.9602	1: 3	15.0	<LOQ
				259.0137			
2-Phenylethyl glucosinolate (gluconasturtiin)	C_15_H_21_NO_9_S_2_	423.0670	422.0597	96.9608	1: 3	20.2	22.0 ± 0.8
				259.0075			mg/kg
							(0.05 mmol/kg)
Methylpentyl glucosinolate	C_13_H_25_NO_9_S_2_	403.0990	402.0915	96.9604	1: 2	24.2	31.7 * ± 2.2
				259.0098			mg/kg
							(0.08 mmol/kg)
Methylpentyl glucosinolate	C_13_H_25_NO_9_S_2_	403,098	402.0911	96.9591	1: 2	24.7	19.5 * ± 1.4
				259.0118			mg/kg
							(0.05 mmol/kg)
Methylpentyl glucosinolate	C_13_H_25_NO_9_S_2_	403,099	402.0913	96.9627	1: 2	25.7	11.5 *± 0.8
							mg/kg
							(0.03 mmol/kg)
Neoglucobrassicin	C_17_H_22_N_2_O_10_S_2_	478.073	477.0657	96.9605	1: 3	26.6	<LOQ
				259.0131			
				446.0503			
Total determined glucosinolates							3740 ± 261 mg/kg
							(9.9 ± 0.7 mmol/kg)

* The results were expressed as sinigrin equivalents.

**Table 2 plants-10-02548-t002:** Results of determined glucosinolates in different protein products from hot-pressed rapeseed cake (HPR) The concentrationsare shown with different colours. The more solid the red colour, the higher the concentration of glucosinolates; the more solid the green colour, the lower is the concentration of glucosinolates.

	HPR-1	HPR-2	HPR-3
Progoitrin	1600 ± 160 mg/kg	1720 ± 10 mg/kg	2565 ± 3.0 mg/kg
Gluconapoliferin	28.3 * ± 0.1 mg/kg	54 * ± 5 mg/kg	3.0 * ± 0.3 mg/kg
Glucoalyssin	<LOD	35.4*± 0.3 mg/kg	52 * ± 0.3 mg/kg
Gluconapin	1479 ± 5.8 mg/kg	2024 ± 2.7 mg/kg	2209 ± 29 mg/kg
Butyl glucosinolate	5.3 * ± 0.05 mg/kg	4.8 * ± 0.03 mg/kg	5.3 * ± 0.1 mg/kg
4-Pentenyl glucosinolate	16.5 * ± 0.08 mg/kg	64.0 *± 1.0 mg/kg	60.9 * ± 0.2 mg/kg
Gluconasturtiin	<LOD	11.3 ± 0.3 mg/kg	28.9 ± 0.4 mg/kg
Neoglucobrassicin	<LOD	1.9 * ± 0.1 mg/kg	10.3 * ± 0.2 mg/kg
3-Methylpentyl glucosinolate	<LOD	1.5 *± 0.05 mg/kg	4.1 *± 0.3 mg/kg
4-Methyltiobutyl glucosinolate	<LOD	<LOD	4.7 * ± 0.3 mg/kg

* The results were expressed as sinigrin equivalents.

**Table 3 plants-10-02548-t003:** Results of determined isothiocyanate and goitrin in different protein products from cold-pressed rapeseed cake (CPR). Limit of detection (LOD) varied from 0.07 mg/kg sample to 0.42 mg/kg sample, whereas the limit of quantification (LOQ) varied from 0.7 mg/kg sample to 4.2 mg/kg sample.

Name of the Isothiocyanate Identified	Spray-Dried Protein Extract	Oven-Dried Protein Extract (Trial CPR-3)	Spray-Dried Product Extract (Trial CPR-4)
	(Trial CPR-2)		
Goitrin	95 ± 7 mg/kg	7 ± 0.5 mg/kg	22 ± 1.5 mg/kg
2-Phenylethyl isothiocyanate	<LOD	<LOQ	<LOD
(from Gluconasturtiin)			
4-Methylthiobutyl isothiocyanate	<LOQ	<LOD	<LOD
3-Butenyl isothiocyanate	<LOD	<LOD	<LOQ
(from Gluconapin)			

**Table 4 plants-10-02548-t004:** The description of different processing parameters that were tested.

Raw Material	Trial No.	Brief Process Description
CPR	1	Soaking overnight, manual liquid–liquid separation and 5 kDa ultrafiltration for concentration, spray drying. 5-day process
CPR	2	Soaking overnight, 0.2 µ microfiltration for liquid–liquid separation and 5 kDa ultrafiltration for concentration, spray drying. 4-day process
CPR	3	Soaking overnight, 0.2 µ microfiltration for liquid–liquid separation and 10 kDa ultrafiltration for concentration, oven drying. 5-day process
CPR	4	Soaking overnight, 0.2 µ microfiltration for liquid–liquid separation and 10 kDa ultrafiltration for concentration, final product in liquid form. 5-day process
HPR	1	Soaking/hydrolysis overnight at 55 °C, enzymatic extraction at pH 8, 0.2 µ microfiltration for liquid–liquid separation and 10 kDa ultrafiltration for concentration, spray drying.
HPR	2	Soaking/hydrolysis overnight at 58 °C, enzymatic extraction at pH 6.7, 0.2 µ microfiltration for liquid–liquid separation and 300 Da nanofiltration for concentration, spray drying.
HPR	3	Soaking/hydrolysis for ca. 4 h at 50 °C, enzymatic extraction at pH 5.0 for 1 h followed by pH adjustment to pH 7.0 a subsequent 30 min reaction time, 0.2 µ microfiltration for liquid–liquid separation and 300 Da nanofiltration for concentration, final product in liquid form.

**Table 5 plants-10-02548-t005:** Dry powder yield of the protein end products CPR-1 to CPR-4 and HPR-1 to HPR-3.

Raw Material	Trial No.	Dry Product Yield
		(g/100 g Dry Raw Material)
CPR	1	4.3
CPR	2	3.6
CPR	3	12.3
CPR	4	6.1
HPR	1	18.1
HPR	2	23.1
HPR	3	14.7

## Data Availability

Not applicable.

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
