# Peer review of "Glucosinolates and Isothiocyantes in Processed Rapeseed Determined by HPLC-DAD-qTOF"

_plants, 2021, doi:10.3390/plants10112548_

Round 1
Reviewer 1 Report
The manuscript is appropriate, the topic is important, the results can fill a gap. But in several places, inattention to spellings and material inaccuracies can be noticed. In addition, there are parts of the experiments that have been presented poorly. I would definitely like to ask these to be improved.
- Title: I don't think this title exactly matches the article. The emphasis is on the comparison of glucosinolates and ITCs, so I think the title should better reflect that.
- Figure 1,3,4: The whole would look better, in a single flowchart where the amounts of glucosinolate and isothiocyanate measured for the different subprocesses are displayed in table form.
- At the various sampling points, it can be seen that both solid and liquid samples were taken. How was the sampling performed?
- What is the dry powder yield? The paper should focus on the GSL and ITC content of the end-product of the proposed procedure.
- Table 1: In the header, M/z should be "[M-H]-".
I don't understand the significance of the 96/97 ratio. What is more, the sulfate group is possible in several other compounds. It would be more appropriate to look for other characteristic and more specific fragments, such as in doi: 10.1186 / s12870-018-1295-4: glucosinolates were identified based on fragments m/z 259.012 and 96.960.
How abundant were the other non-quantified compounds for total glucosinolate levels? Is it worth expressing in equivalents or just write down the AUC values of the measured molecules?
- Figure 2: If there are standards, why are they not shown in the figure? The figure has limited usefulness in its current form, the tenth molecule is missing (I think it is 4-methyltiobutyl glucosinolate). The colour scale does not contain a value for compounds below the LOD. I also could not understand your scale: Why not map to raw abudance data?
- In the HRPs and CRPs more than one parameter at the same time was changed, plus we don't know anything about the different filtration technologies. Please add some information about the filtering methods. Was vacuum used? Has there been a measured ITC loss? Do you filter particulate matter or also protein subfractions? HPR-3 also occurred at two types of pH, yet it is considered a single method. Why isn't it separated? Are replicates technical replicates or the different workflow types were repeated several times?
- Table S1: If the measured glucosinolates are here, add theoretical mass, measured monoisotopic mass, and error (ppm) to these tables.
Table S2, S3, S4, S5, S7: I could suggest merging these tables into one, and the results of CRP GLS and ITC could also be included. The "m / z" column should be "[M-H]-".
Table S6: I think it is irrelevant to list the same metabolite from each sample in a table.
- Line 40-41: CN group instead of SCN?
- Line 177: ...(abundance in MS 1403 ± 160)... If it is important, why were not there the values of the other molecules too? Add units of measure.
- Lines 191-193: at 70 ° C the enzyme mirosinase is not completely inactivated according to the literature (doi: 10.1016 / j.plaphy.2005.03.015).
- Line 295-296: What results or literature data do you base this on?
- Line 336-337: How much heat is produced? Has it been measured? Can you give an estimate value?
- Line 364-366: With lead acetate or barium acetate, is it sure that GLS does not precipitate causing a negative measurement error? Have measurements been made in this direction? Add data or references.
- Line 392-393: Why are the resultes of the calibration not included in at least the supplement?
- In general: Species names must be in italics. Please correct the spelling mistakes and check and standardize punctuation.
Reviewer 2 Report
Minor comments.
Title. Possible change to ..... in processed rapeseed...
Graphical abstract. Review text "Analysis of glucosinolates and their hydrolysis products"
Results.
Glucosinolates identification. The fragment of 96/97 ratio is interesting for the identification, but why the marker fragment for glucosinolates, the 259 m/z, was not also used as backup/confirmation for identification?
The NGB/MGB (477 m/z) why is not separated in the analysis? MGB elutes earlier and NGB can be diagnosed also by the MS2 presence of 447 m/z, in the used conditions this separation was not possible? (I doubt it).
Progoitrin in HPR (Table 2), >1.6 - 2.5 mg/g, how toxic is the presence of progoitrin at this concentration? What's the potential for goiter problems with such a high content? - the cakes are supposed to be used as feedstuff? It is possible to be accumulated if continuous intake is avaialable, and therefore, how dangerous/potentially toxic this can be?
Extraction of glucosinolates.
70% aqueous methanol OK, there is no heating and a minimum of 20 min. at 70 °C to inhibit myrosinase. Only 5 min. extraction. Probably the concentraiton of glucosinolates is understimated and without inhibition of myrosinase, the extraction of ITCs only for 5 min. is also not enough.
The data presented in terms of content of glucosinolates many not be as correct/accurate as expected. Did you optimize the extraction under this conditions in a different publication? is it available?
Round 2
Reviewer 1 Report
Dear Author!
I would like to thank the authors for the answers. And thank you for the detailed addition. I have only found a few small errors in the manuscript that need to be corrected, but I have no other objections to the material.
1. Table1: Methylpentyl glucosinolate 2,3 row, [M-H]- column: punctuation error.
2. Figure 1: In the head of the table: I think this are Glucosinolates and Isothiocyanates not Glucosinolates and Goitrin.
3. Line 567-568: ...In this trial, goitrin
had twice the concentration compared to Trial CPR – 1 (Figure 1). ... This is not shown is Figure 1.
